# Infrared Image Deconvolution Considering Fixed Pattern Noise

**DOI:** 10.3390/s23063033

**Published:** 2023-03-11

**Authors:** Haegeun Lee, Moon Gi Kang

**Affiliations:** School of Electrical and Electronic Engineering, Yonsei University, Seoul 03722, Republic of Korea

**Keywords:** infrared image, non-blind deconvolution, fixed pattern noise, non-uniformity correction, regularization, optimization

## Abstract

As the demand for thermal information increases in industrial fields, numerous studies have focused on enhancing the quality of infrared images. Previous studies have attempted to independently overcome one of the two main degradations of infrared images, fixed pattern noise (FPN) and blurring artifacts, neglecting the other problems, to reduce the complexity of the problems. However, this is infeasible for real-world infrared images, where two degradations coexist and influence each other. Herein, we propose an infrared image deconvolution algorithm that jointly considers FPN and blurring artifacts in a single framework. First, an infrared linear degradation model that incorporates a series of degradations of the thermal information acquisition system is derived. Subsequently, based on the investigation of the visual characteristics of the column FPN, a strategy to precisely estimate FPN components is developed, even in the presence of random noise. Finally, a non-blind image deconvolution scheme is proposed by analyzing the distinctive gradient statistics of infrared images compared with those of visible-band images. The superiority of the proposed algorithm is experimentally verified by removing both artifacts. Based on the results, the derived infrared image deconvolution framework successfully reflects a real infrared imaging system.

## 1. Introduction

Infrared images enable the detection of subjects, even in poor image acquisition conditions, such as extremely dark illumination or bad weather, by sensing the thermal radiation of each object [1]. Owing to its unique characteristics, which are different from those of visible-band images, infrared imagery has been extensively utilized in various fields of application, such as military, surveillance, medical science, agriculture, and fire detection [2]. The use of a focal plane array (FPA) has enabled the mass production of infrared cameras and accelerated the advancement in thermal imaging systems [3]. However, the obtained thermal images generally suffer from two major degradations: fixed pattern noise (FPN) and blurring artifacts. Thus, numerous signal-processing-based algorithms have been studied to effectively reduce the observed limitations and expedite the efficient utilization of these imaging sensors by restoring precise thermal information.

FPN, widely known as non-uniformity (NU) noise, is spatially patterned noise that is primarily generated by manufacturing flaws, inhomogeneous responsivities of pixels, and dark currents of photo detectors [4]. Complementary metal–oxide semiconductor (CMOS) image sensors for infrared imaging systems suffer from higher non-uniformity than charge-coupled device (CCD) sensors, owing to the presence of amplifiers. In particular, column FPN, which denotes strongly column-directed spatial noise, is the most-commonly observed limitation owing to the presence of column-parallel amplifiers and the analog-to-digital converter (ADC) of each column. As the column FPN occupies a large proportion of the degradation factor for the quality of thermal images, we focused on effectively reducing the deterioration while preserving the other image contents.

The other degradation factor, image blur, fades the sharp subjects in the observed images owing to some physical limitations of the image acquisition process, which is represented by the point spread function expressed by the acquired observation of a single-point subject. Artifacts are divided into two types depending on the problems: motion blur generated by camera shake or the motion of objects and optical blur caused by the optical system or lens–camera arrangement [5]. Image deconvolution removes the observed blurs by solving the inverse problem of linear degradation models, and various approaches have been studied for visible-band images [6,7,8]. In particular, non-blind deconvolution algorithms can restore the observed degradations when the point spread function is given and have been extensively studied for the deep consideration of imaging systems considering restored image characteristics [9,10,11]. Several studies have been conducted to overcome blurs and random noise for infrared images [12,13]; however, they did not consider FPN in the modeling, which dominates the quality of the observed infrared images. Individual restored images of previous studies illustrated fine restoration results under the random noise assumption, but the earlier-proposed algorithms are difficult to employ practically because of the presence of FPN in real infrared images.

In this paper, we propose a non-blind deconvolution algorithm for infrared images that considers an FPN. First, we derived an observation model for thermal images that jointly considers blur, random noise, and FPN. As both the FPN and blur models were linearly designed in previous studies, they were combined in the lexicographically ordered vector representations. Subsequently, the objective function for estimating the column FPN was derived in the maximum a posteriori (MAP) framework. Since the column FPN primarily consists of column-directed components, a regularization function that reflects the characteristics is proposed for precise estimation. Subsequently, an infrared image deconvolution strategy was developed based on the characteristics of infrared images that are different from those of visible-band images. The gradient distributions of the infrared images were compared with those of conventional images to investigate their statistical characteristics. Finally, efficient minimization methods for each optimization strategy were derived: the alternating directional-method-of-multipliers (ADMM)-based method for global gradient priors of FPN and the iteratively reweighted-least-squares (IRLS)-based algorithm for the spatially variant data fidelity of deconvolution problems. Consequently, the observed degradations of the infrared images were more effectively reduced by the proposed algorithm compared to conventional algorithms, which successively remove observed FPN and blur artifacts, as shown in Figure 1: the residual noise of the FPN-removed image is boosted during the deconvolution process in the conventional framework, while both artifacts are successfully suppressed in the proposed framework.

The remainder of this paper is organized as follows: In Section 2, we investigate previous studies on FPN and image deconvolution problems and derive an appropriate observation model for infrared images. In Section 3, we propose an infrared image deconvolution algorithm that jointly considers the FPN and the observed blurs. In Section 4, the performance of the proposed algorithm is compared with that of the conventional algorithms in terms of denoising and deconvolution. Finally, in Section 5, we present our conclusions.

## 2. Related Work

### 2.1. Fixed Pattern Noise of Thermal Images

FPN refers to the spatial noise that is usually generated by the hardware characteristics of thermal imaging systems. It is composed of photo response non-uniformity (PRNU) and dark signal non-uniformity (DSNU), which are commonly caused by differences in pixel responsivity. The former denotes the gain of observations versus the true temperatures of the subjects, and the latter represents the offset that is independent of them. The observed image containing the FPN of the infrared images was modeled using the following equation:(1)y(i,j)=a(i,j)·x(i,j)+b(i,j)+n(i,j),
where y(i,j) and x(i,j) represent the $\left(col\cdot (i-1)+j\right)$ elements of the observed image and the latent image, respectively. n(·) represents random noise, which is generally assumed to be distributed with a Gaussian distribution. Moreover, a(·) and b(·) represent the characteristics of FPN: the multiplicative term, PRNU, and the additive term, DSNU. The goal of the FPN removal algorithms is to estimate the true values of a(·) and b(·) and eliminate them from the observations.

Non-uniformity correction (NUC) compensates for the inherent parameters of imaging detectors to remove the observed FPN. NUC has been studied in two ways: scene-based and calibration-based methods. Scene-based methods utilize multiple raw images from the same infrared camera and some statistical assumptions regarding the observed FPN to remove the unknown NU [15,16,17]. Even though they require rigid input data, such as sequences of infrared images, they commonly show lower accuracy with larger complexities compared to calibration-based methods. Calibration-based algorithms use infrared data of a specific temperature reference generated by a blackbody radiation source [14,18,19,20]. Two-point methods [18] estimate the gain and offset of the linearly approximated FPN models based on two observations of high and low temperatures. Multi-point algorithms [19] extend the aforementioned study [18] and demonstrate improvements in performance by modeling the piecewise linearity of the observed FPNs more precisely. Additionally, curve-fitting methods [14] have been studied to generalize the non-linear characteristics of responsivities and effectively operate for thermal images covering a wide range of temperatures. Such calibration-based algorithms efficiently remove the observed FPN and enable the real-time implementation of NUC; however, they neglect the random noise in the observations, which induces critical side-effects in the restored image qualities. In this study, we propose a calibration-based FPN estimation method in an optimization framework to effectively reflect both noises in deconvolution problems with reasonable computations.

### 2.2. Image Deconvolution

Image deconvolution restores high-quality images from degraded observations by removing degradations such as blurs and random noise. The degradation process of a common image acquisition process is defined as follows:(2)y=Hx+n,
where y, x, and n denote lexicographically ordered vector representations of y(i,j), x(i,j), and n(i,j), respectively. H represents the system matrix of the blur kernel h, and the deconvolution algorithm is divided into non-blind and blind, depending on whether this term is known or unknown. In this study, we attempted to solve non-blind deconvolution problems for infrared images, which are still highly ill-posed owing to the singularity of H and the irregularity of n. In addition to blur degradation, infrared images commonly suffer from the aforementioned FPN, as depicted in Figure 2. Therefore, the infrared image observation model including blurs and FPN is derived as follows:(3)y=AHx+b+n,
where b represents the vectorized expression of additive FPN b(i,j) and A denotes the diagonal matrix, whose $\left(col\cdot (i-1)+j\right)$th diagonal component is multiplicative FPN a(i,j). High-quality infrared images can be obtained by solving the inverse problem of Equation (Equation 3), jointly removing the blurs and FPN.

The objective function of the infrared image restoration problem is defined as follows:(4)$$F(x)=\frac{1}{2}||\mathrm{AHx}+b-{y||}_{2}^{2}+\lambda \cdot R\left(x\right),$$
where the first term represents the data fidelity, which penalizes the restored image to the observation model, and the second term, R(x), represents the regularization function, which reduces the uncertainty of inverse problems by employing prior information on high-quality infrared images. Moreover, λ symbolizes the regularization parameter that controls the significance between the data fidelity and the regularization function, which primarily determines the restored image characteristics. Therefore, to enhance the image quality restored by solving this problem, a regularization function that derives the unique characteristics of high-quality thermal images should be designed.

Various regularization approaches have been studied to accurately describe the innate characteristics of conventional visible-band images. Hunt et al. [21] derived the constrained least-squares approach to characterize the smoothness of high-quality images, which is also known as the smoothness prior. Rudin et al. [22] proposed total variation regularization to preserve edges in the horizontal and vertical directions, and Farsiu et al. [23] extended this idea using bilateral filters to jointly consider diagonal edges. Russel et al. [24] first invented the natural image prior derivation and considered the statistical characteristics of natural images. Moreover, various studies have attempted to precisely model them using various distributional models. Cho et al. [25] attempted to describe different patchwise statistical attributes by deriving a spatially variant regularization function for the generalized Gaussian distribution (GGD) model. Zoran et al. [26] employed the Gaussian mixture model (GMM) and derived the expected patch log likelihood (EPLL) framework to consider patch statistics more deliberately. Lee et al. [10] proposed an automated prior selection algorithm that precisely estimates the inherent gradient statistics from the observations. However, few studies have been conducted on infrared images in this field. Thus, we propose an effective regularization function by analyzing the infrared image statistics.

## 3. Proposed Algorithm

In this paper, we propose an infrared image deconvolution algorithm that simultaneously overcomes both blur degradation and FPN. First, the multiplicative and additive components of the FPN were calculated from the calibration data based on the visible features of the column FPN. Subsequently, a non-blind image deconvolution problem that incorporates the estimated FPN was derived, reflecting infrared image statistics within the maximum a posteriori (MAP) framework. Finally, the optimization strategies for FPN estimation and image deconvolution were separately studied for each process to enhance the efficiency depending on the characteristics of the objective functions.

### 3.1. Estimation of Fixed Pattern Noise

Calibration-based FPN estimation algorithms commonly require pre-processing tasks to extract the physical characteristics of image acquisition devices, but have been extensively employed for real-world cameras owing to their high efficiency. The method utilizes calibration data that capture the blackbody at constant temperatures and estimates the additive and multiplicative terms of FPN from the observations, as shown in Figure 3a. The image $y_{k}$ of a blackbody at the same temperature $t_{k}$ can be modeled as follows:(5)$$y_{k}=t_{k}a+b+n,$$
for 1≤k≤K, where *K* indicates the number of calibration frames, and a represents the vector representation of a(i,j). In the experiments, $t_{k}$ was determined as the mean value of the measured blackbody to maintain the average intensity of the infrared images after the FPN removal process. The FPN estimation problem corresponds to the a and b-determination problems considered in this study. Previous FPN removal algorithms also aimed to accurately find the components, but failed to consider the presence of random noise n in the reference data, which resulted in strong errors in the estimated FPN.

We interpreted the FPN estimation problem in the optimization framework to reduce side-effects and improve the estimation performance by reflecting the characteristics of the observed FPN. In the proposed derivation, the multiplicative noise a is calculated by subtracting the consecutive calibration frames in Equation (Equation 5), the optimization problem of which is expressed as follows:(6)$$\hat{a}=\underset{a}{arg min}\left\{\frac{1}{2}\sum_{k=1}^{K-1}\left|\right|\left(t_{k+1}-t_{k}\right)a-\left(y_{k+1}-y_{k}\right){\left|\right|}_{2}^{2}\right\}.$$

Subsequently, the additive noise b was computed by determining the least-squares solution using the estimated a:(7)$$\hat{b}=\underset{b}{arg min}\left\{\frac{1}{2}\sum_{k=1}^{K}||b+t_{k}a-y_{k}{\left|\right|}_{2}^{2}\right\}.$$

The derived Equations (Equation 6) and (Equation 7) compensate for the boosted random noise problem by reducing randomness with the use of multiple frame data. Therefore, the accuracy of the FPN estimation process depends significantly on the number of reference frames, and strong random noise is difficult to remove using this stochastic approach.

To overcome these practical limitations, we developed a regularization strategy that reduces the ill-posedness of the problem by introducing prior information to the observed column FPN. The column FPN primarily consists of column-directed patterns without row-directed components and exhibits extremely high sparsity only in their vertical gradients with the exception of horizontal gradients. Thus, the vertical edge-preserving prior is derived by enforcing gradient sparsity in the column direction, and the regularized problem including this information is developed as follows:(8)$$F_{a}\left(a\right)=\sum_{k=1}^{K-1}\frac{\lambda_{a,k}}{2}\cdot \left|\right|\left(t_{k+1}-t_{k}\right)a-\left(y_{k+1}-y_{k}\right){\left|\right|}_{2}^{2}+\left|\right|\nabla_{c}a{\left|\right|}_{1}^{1},$$
and
(9)$$F_{b}\left(b\right)=\sum_{k=1}^{K}\frac{\lambda_{b,k}}{2}\cdot ||b+t_{k}a-y_{k}{\left|\right|}_{2}^{2}+\left|\right|\nabla_{c}b{\left|\right|}_{1}^{1},$$
where $F_{a}\left(a\right)$ and $F_{b}\left(b\right)$ represent the derived objective functions and $\nabla_{c}$ denotes the gradient operator in the column direction. The penalized column gradient sparsity was expected to strongly suppress the observed random noise while reconstructing similar values in the column direction for the computed FPN.The estimated FPN components demonstrate high precision by closely describing the attributes of the column FPN, regardless of the observed infrared images, as depicted in Figure 3b.

### 3.2. Thermal Image Deconvolution

The qualities of the restored images are primarily determined by how accurately and deliberately the regularization function describes the characteristics of the high-quality images. Therefore, for the fine restoration of thermal images, prior information on clean thermal images should be investigated in advance. Compared with high-quality images, thermal images generally exhibit fewer high-frequency components, such as details and textures, but smoother areas, such as flat regions. These distinctive visual characteristics are displayed as gradient distributions of different shapes for various RGB–thermal image datasets [13,27,28,29], as depicted in Figure 4.

In this study, we analyzed the gradient statistics to characterize the high-quality thermal images. Gradient priors have been extensively studied for visible-band images owing to their effectiveness and computational simplicity [9]. The first-order derivative operator, which computes the gradient statistics of observations, filters out the high-frequency information of the images. Thus, the output signals of visible-band images are generally distributed as heavy-tailed distributions, owing to the large number of sharp edges. In contrast, fewer details and a wider smoothness of thermal images are exhibited as higher peaks and shorter tails in their gradient distributions compared to those of visible-band images. We investigated these observable differences by statistically modeling the distributions using the following GGD model *f*:(10)$$f\left(\nabla x;p,\sigma \right)=\frac{p}{2\sigma \Gamma (1/p)}\exp\left(-\frac{{\left||\nabla x|\right|}^{p}}{\sigma }\right),$$
where Γ(·) represents the gamma function and *p* and σ denote two shape parameters of the GGD that determine the height of the peaks and the width of the tails of the distributions, respectively.

We employed statistical modeling in our previous study [10], which numerically estimated the two parameters that most closely describe the reference distributions. As shown in Figure 5, the comparatively higher peaks and shorter tails of the thermal images are represented by smaller values of *p* and σ. To use these statistics as prior information, the model estimated in Equation (Equation 10) can be transformed as follows:(11)$$\log\left(f(\nabla x;p,\sigma )\right)\propto -\frac{{\left||\nabla x|\right|}^{p}}{\sigma }.$$

In the MAP framework of image deconvolution problems, the thermal image prior in Equation (Equation 11) is derived as the regularization function of the objective function F(x), as follows:(12)$$F(x)=\frac{1}{2}||\mathrm{AHx}+b-{y||}_{2}^{2}+\lambda \cdot {\left||\nabla x|\right|}_{p}^{p},$$
where the regularization parameter λ is inversely proportional to σ. High-quality thermal images can be restored by solving the above image deconvolution problem for small values of *p* and large values of λ, reflecting the strong gradient sparsity of thermal image statistics.

### 3.3. Optimization Methods

In this subsection, we derive optimization algorithms to minimize the proposed objective functions in Equations (Equation 8), (Equation 9), and (Equation 12). The components of the FPN, the multiplicative FPN a and the additive FPN b, were successively estimated in advance, and the observation model of infrared images in Equation (Equation 3) was obtained. Subsequently, the clean infrared image was restored by solving the deconvolution problem. The optimization strategy was studied differently for each variable to efficiently solve problems depending on their attributes.

First, the convex and non-smooth problem of multiplicative FPN a in Equation (Equation 8) was solved using the ADMM framework. The ADMM is a powerful optimization algorithm because of its fast convergence rate in overcoming the non-smoothness of the objective functions. The strategy efficiently solves complicated problems by simplifying them with additional auxiliary variables and splitting them into several sub-problems. The derived objective function is expressed as follows:(13)$$\begin{array}{cc} L_{a}\left(a,u_{a},v_{a}\right)= & \frac{1}{2}\sum_{k=1}^{K-1}\lambda_{a,k}\cdot \left|\right|\left(t_{k+1}-t_{k}\right)a-\left(y_{k+1}-y_{k}\right){\left|\right|}_{2}^{2}+\left|\right|u_{a}{\left|\right|}_{1}^{1}\\ & +\frac{\rho_{a}}{2}\left|\right|u_{a}-\nabla_{c}a+v_{a}{\left|\right|}_{2}^{2}-\frac{\rho_{a}}{2}\left|\right|v_{a}{\left|\right|}_{2}^{2}, \end{array}$$
where $u_{a}$ denotes an auxiliary variable that replaces $\nabla_{c}a$ and $\rho_{a}$ indicates a penalty parameter that enforces the similarity between $\nabla_{c}a$ and $u_{a}$. Moreover, $v_{a}$ denotes a dual variable in the ADMM derivation. The objective function was minimized by successively solving the problems of variables a, $u_{a}$, and $v_{a}$ in the ADMM framework.

The a sub-problem is characterized by smoothness and convexity; therefore, the inverse problem can be efficiently solved in the frequency domain as follows:(14)$$\begin{array}{cc} a^{n+1} & =\underset{a}{arg min}\left\{\frac{1}{2}\sum_{k=1}^{K-1}\lambda_{a,k}\cdot \left|\right|\left(t_{k+1}-t_{k}\right)a-\left(y_{k+1}-y_{k}\right){\left|\right|}_{2}^{2}+\frac{\rho_{a}}{2}\left|\right|\nabla_{c}a-u_{a}^{n}-v_{a}^{n}{\left|\right|}_{2}^{2}\right\}\\ & =F^{-1}\left\{\frac{F\left\{\sum_{k=1}^{K-1}\lambda_{a,k}\left(t_{k+1}-t_{k}\right)\cdot \left(y_{k+1}-y_{k}\right)+\rho_{a}\nabla_{c}^{T}\left(u_{a}^{n}+v_{a}^{n}\right)\right\}}{|F\left\{\sum_{k=1}^{K-1}\lambda_{a,k}\left(t_{k+1}-t_{k}\right)\cdot I\right\}{|}^{2}+\rho_{a}|F\left\{\nabla_{c}\right\}{|}^{2}}\right\}, \end{array}$$
where F and $F^{-1}$ denote the fast Fourier transform (FFT) and inverse fast fast Fourier transform (IFFT) operations, respectively. Since the complex problem is easily optimized by simple division in the frequency domain, it significantly reduces the computations and determines the efficiency of the algorithm. The convex, but non-smooth $u_{a}$ sub-problem is expressed as follows:(15)$$u_{a}^{n+1}=\underset{u_{a}}{arg min}\left\{\frac{\rho_{a}}{2}\left|\right|u_{a}-\nabla_{c}a+v_{a}^{n}{\left|\right|}_{2}^{2}+\left|\right|u_{a}{\left|\right|}_{1}^{1}\right\},$$
where the gradient operator in the regularization term in Equation (Equation 8), which incurs large computational costs in optimization, is abbreviated. Therefore, this problem can be solved using the following soft thresholding method:(16)$$\begin{array}{cc} u_{a}^{n+1} & =shirink(\nabla_{c}a-v_{a}^{n},1/\rho_{a})\\ & =\max\{|\nabla_{c}a-v_{a}^{n}\left|-1/\rho_{a},0\right\}\cdot sign\left(\nabla_{c}a-v_{a}^{n}\right), \end{array}$$
where shirink(·) stands for the shrinkage operator and sign(·) denotes the signum function that returns the signs of the real vectors. The dual variable $v_{a}$ is then updated as follows:(17)$$v_{a}^{n+1}=v_{a}^{n}+u_{a}^{n+1}-\nabla a^{n+1}.$$

Upon calculating the multiplicative FPN a using Equations (Equation 14), (Equation 16), and (Equation 17), the additive FPN b is computed similarly in the ADMM framework. The objective function of b in Equation (Equation 9) is characterized by convexity and non-smoothness, but shows less complexity compared to that of a owing to the absence of coefficients $t_{k}$. The process requires a previously estimated a and aims to minimize the following optimization problem:(18)$$\begin{array}{cc} L_{b}\left(b,u_{b},v_{b};a\right) & =\frac{1}{2}\sum_{k=1}^{K}\lambda_{b,k}\cdot ||b+t_{k}a-y_{k}{\left|\right|}_{2}^{2}+\left|\right|u_{b}{\left|\right|}_{1}^{1}\\ & +\frac{\rho_{b}}{2}\left|\right|u_{b}-\nabla_{c}b+v_{b}{\left|\right|}_{2}^{2}-\frac{\rho_{b}}{2}\left|\right|v_{b}{\left|\right|}_{2}^{2}, \end{array}$$
where $u_{b}=\nabla_{c}b$ and $v_{b}$ are auxiliary variables for reducing the complexity and $\rho_{b}$ penalizes the fidelity of $u_{b}$ to $\nabla_{c}b$. From the derived function, the convex and smooth b sub-problems are efficiently solved in the frequency domain as follows:(19)$$\begin{array}{cc} b^{n+1} & =\underset{b}{arg min}\left\{\frac{1}{2}\sum_{k=1}^{K}\lambda_{b,k}\cdot ||b+t_{k}a-y_{k}{\left|\right|}_{2}^{2}+\frac{\rho_{b}}{2}\left|\right|\nabla_{c}b-u_{b}^{n}-v_{b}^{n}{\left|\right|}_{2}^{2}\right\}\\ & =F^{-1}\left\{\frac{F\left\{\sum_{k=1}^{K}\lambda_{b,k}\cdot y_{k}+\rho_{b}\nabla_{c}^{T}\left(u_{b}^{n}+v_{b}^{n}\right)\right\}}{|F\left\{\sum_{k=1}^{K}\lambda_{b,k}\cdot I\right\}{|}^{2}+\rho_{b}|F\left\{\nabla_{c}\right\}{|}^{2}}\right\}. \end{array}$$Subsequently, the other variables $u_{b}$ and $v_{b}$ are computed using calculations similar to $u_{a}$ and $v_{a}$ in Equations (Equation 16) and (Equation 17). Through the above calculations, the multiplicative FPN a and additive b are efficiently achieved, representing the visual characteristics of the column FPN.

Finally, the infrared image deconvolution problem in Equation (Equation 12) is developed based on the estimated FPN a and b. However, because the objective function is composed of a spatially variant data fidelity term and a non-convex regularization term, it is difficult to minimize and requires a large number of computations for convergence. Although the ADMM performs well in solving some optimization problems, it fails to treat spatial variance, which cannot be interpreted in the frequency domain. Therefore, we introduced an iteratively reweighted least-squares (IRLS) strategy to simplify the complexity by overcoming the non-convexity of the proposed $L_{p}$-regularized problem. The IRLS approximates the non-convex term to a quadratic expression by computing the coefficients of the corresponding term regarding the variables of the current iteration, *n*, as those of the next iteration, n+1, and derives a series of weighted least-squares problems. For the function proposed in Equation (Equation 12), the approximated convex function can be expressed as follows:(20)$$\begin{array}{c} F(x^{n+1})\approx \frac{1}{2}\left|\right|{\mathrm{AHx}}^{n+1}+b-y{\left|\right|}_{2}^{2}+\lambda \cdot {\left(\nabla x^{n+1}\right)}^{T}W\left(\nabla x^{n+1}\right), \end{array}$$
where
(21)$$W=diag\left(p\cdot {\left({\left(\nabla x{\left(i,j\right)}^{n}\right)}^{2}+\epsilon \right)}^{\frac{p}{2}-1}\right),$$
ϵ>0 is a negligibly small parameter, and $\nabla x{\left(i,j\right)}^{n}$ represents the (i,j)-th components of the vector $\nabla x^{n}$ at iteration *n*. The optimization problem of the compressive sensing field has been simply solved by direct inversion; however, it cannot be applied in this deconvolution field owing to the spatially variant variant multiplicative noise matrix A and the presence of gradient operator ∇. Therefore, we employed the conjugate gradient method, which efficiently solves complex, but convex problems, even with spatially variant operators, through numerical iterations. Consequently, a high-quality thermal image x can be restored.

## 4. Experiment Results

In this section, we validated the performance of the proposed algorithm in terms of denoising and deconvolution. The former handles the problems of removing FPN and random noise from infrared images, whereas the latter jointly considers blur artifacts in addition to noise. Both problems commonly aim to restore high-quality infrared images from the observed poorly conditioned images.

The synthesized versions of the high-quality infrared images of the CATS dataset [28] shown in Figure 6 and real-world infrared images of the SRIP dataset [13] were utilized to evaluate the restoration performance. For noise removal problems, the clean thermal images were contaminated by column-directed FPN and additive white Gaussian noise at the 20 dB scale, and they were formerly degraded by the point spread functions of real blurs [30] for the deconvolution problems. On the other hand, the observed real thermal images were utilized without any additional degradation.

### 4.1. FPN Removal

In this section, we evaluated the efficacy of the proposed algorithm in simultaneously reducing FPN and random noise. The proposed FPN removal strategy corresponds to the calibration-based method, which requires observed data of ground-truth temperatures, so the calibration data of blackbodies with four identical temperatures were additionally utilized. For a fair comparison, the restoration performance of the proposed algorithm in overcoming noise was compared with well-known calibration-based FPN removal algorithms: the two-point [18], multi-point [19], and polynomial [14] methods.

Previous studies focused on accurately estimating multiplicative and additive noise from calibration data by assuming a linear observation model. Therefore, we demonstrated the results of the proposed algorithm in two ways: after estimating the FPN components using our FPN calculation algorithm, one method (Proposed 1) obtains the restored image from the linear calculation, similar to the two-point method [18], and the other (Proposed 2) employs the proposed image deconvolution algorithm in Equation (Equation 12), assuming H is an identical matrix, to only consider denoising problems.

The restored infrared images of the noise removal algorithms are qualitatively compared in Figure 7. As illustrated, most algorithms successfully recovered identifiable contents from the degraded observations, and Proposed 1 demonstrated a more effective denoising performance than conventional algorithms by accurately estimating the elements of the column FPN. This is because the common assumption of most previous studies was that calibration images are noise-free. However, they were also obtained by a thermal imaging system that contains random noise during the image acquisition process, and this is effectively considered in the proposed FPN estimation algorithm through the regularization function specifying the column FPN in Equations (Equation 8) and (Equation 9), respectively. Even though the multiplicative and additive FPN terms were precisely estimated, there still remained strong residual noise in the resulting images of Proposed 1, as the linear model approach aimed to only remove the observed FPN, not random noise. In the proposed optimization framework, the observed noise is suppressed by the statistical characteristics of high-quality thermal images, as shown in the results of Proposed 2, which illustrated the closest image characteristics to the original infrared image, both in the flat regions of the parking lot and the detailed regions of cars.

To quantitatively compare the restored image qualities, four image quality assessments (IQAs) were employed: peak-signal-to-noise ratio (PSNR), structural similarity index map (SSIM) [31], roughness (Ro) [32] index, and effective roughness (ERo) [33] index. The PSNR and SSIM are the most-widely utilized IQAs to quantify the restoration performance by measuring the similarity of the restored images compared to the corresponding original image. Therefore, they commonly require reference images that are nearly impossible to be acquired for real-world images. Ro quantifies the degree of NU, which is the most-critical artifact in thermal images, by computing the ratio between the energies of an image and its edges, and ERo extends the metric utilizing high-frequency components of individual images. These two measurements are also available to measure the restoration performance for real-world infrared images because they do not require the original images. A comparison of the measured assessments is given in Table 1. As illustrated, the performance of the FPN removal algorithms became more effective as the FPN components were more delicately modeled with the two-point, multi-point, polynomial, and Proposed 1, displaying higher values in the PSNR and SSIM and lower values in Ro and ERo. Moreover, Proposed 2 outperformed the other algorithms by a large margin for most images and measurements, verifying its unrivaled performance when considering random noise.

### 4.2. Infrared Image Deconvolution

The performance of the proposed non-blind thermal image deconvolution algorithm was experimentally validated and compared with those of conventional studies on visible-band images [6,8,10,11]. To construct the simulation set, clean infrared images were blurred by real image blurs [30] and subsequently degraded by the column FPN and random noise. The proposed algorithm simultaneously overcomes the observed blurs, FPN, and random noise by jointly considering the degradations. However, conventional deconvolution approaches aim to only remove blurs and random noise. Therefore, for conventional algorithms, the observed FPN of the degraded images is first reduced by the calibration-based NUC method [14], and subsequently, the non-blind deconvolution problem is solved to remove the blur degradations. On the other hand, the proposed algorithm was operated simultaneously to restore high-quality thermal images from degraded observations. The blur kernel information is provided for both approaches, as they commonly solve non-blind deconvolution problems.

There have been few studies on infrared image deconvolution problems, as thermal images are typically employed for the detection of subjects, rather than the observation of individual content. However, as the application fields of these techniques have broadened, the demand for high-precision thermal information has increased, and their physical limitations can be overcome using deconvolution algorithms, which have not been extensively studied for infrared images. Thus, we employed state-of-the-art non-blind image deconvolution methods for visible-band images to qualitatively validate the proposed infrared image restoration algorithm. Levin et al. [6] derived a probabilistic model expression for deblurring problems and defined the natural image prior, which is a common characteristic of high-quality visible-band images. Liming et al. [8] employed an iteratively reweighted $L_{1}$ minimization algorithm to overcome the non-convexity in the gradient sparsity of deconvolution problems. Jon et al. [11] focused on the distributions of overlapping patches and combined overlapping group sparsity with a natural image prior to penalize grouped sparsity. Lee et al. [10] proposed an automatic prior selection algorithm that automatically determines the most-effective prior term for visible-band images. The thermal images restored by these well-established algorithms for visible-band images were compared with those of the proposed algorithm.

Figure 8 shows the resulting thermal images restored by the individual deconvolution algorithms. The degraded image was initially denoised by the polynomial-based FPN removal algorithm. However, as illustrated in the figure, it still suffered from considerable noise in flat regions and blurring artifacts in detailed regions. The results of deconvolution algorithms commonly demonstrate effective performance in recovering high-frequency information, such as edges and details in the restored images, but most of the severe side-effects are commonly observed in the form of boosted noise; in particular, Liming et al. [8] and Jon et al. [11] transformed the residual noise into overly enhanced patterns in flat regions. Meanwhile, the restored images of the proposed algorithm successfully improved the detailed regions while effectively suppressing noise. The restored thermal images are quantitatively compared in Table 2. The results of the proposed algorithm demonstrated better performance on most images in terms of the PSNR and SSIM, demonstrating that the restored images closely describe the corresponding original images. In particular, the latter showed much more overwhelming performance, which were quite more sensitive to noise than the former. Additionally, the Ro and ERo of the proposed algorithm mostly exhibited lower values than those of the conventional algorithms, verifying that NU was effectively removed during the deconvolution process of the proposed framework.

Finally, the real-world images of the SRIP dataset were restored using the aforementioned FPN removal and non-blind deconvolution framework. As mentioned above, only the proposed algorithm jointly considers degradation and restores the true thermal information from the observations. As the images of the dataset contain real-world blurs, FPN, and random noise, no other degradation processes were conducted on them. The blur kernels, which should be known for non-blind image deconvolution algorithms, are initially reconstructed from the captured line spread function of the infrared camera. Figure 9 illustrates the observed real data of visible-band and infrared images and visually compares the qualities of the restored infrared images. As illustrated, infrared images are commonly contaminated by an amount of FPN and random noise with smoothing artifacts compared to visible-band images showing a lack of details. Even after the initial FPN removal algorithm, there remained much noise owing to the harsh conditions of the real image acquisition process compared to that of synthetic images. The purpose of the proposed algorithm was to improve the details of infrared images while removing unwanted artifacts. Consecutive FPN removal and deconvolution processes for conventional algorithms commonly fail to obtain high-quality images from observed infrared data. Levin et al. [6] and Lee et al. [10] reduced some of the observed noise, but failed to preserve the detailed information of each image. Liming et al. [8] and Jon et al. [11] sharply enhanced high-frequency information, but the noise was also boosted during the restoration process. In contrast, the proposed algorithm successfully recovered the edge components while effectively removing both the FPN and random noise. The performance of the proposed algorithm in NUC was also validated by the measurements in Table 3: the Ro and ERo of the proposed algorithm demonstrated the lowest values among those of the other conventional algorithms.

## 5. Conclusions

Although actual thermal image acquisition systems include a series of blur and FPN problems, they have been independently studied in previous studies for simplification. In this study, we proposed an infrared image deconvolution algorithm that simultaneously overcomes both degradations by jointly considering them using the derived observation model. The visual features of a column FPN were investigated and utilized to specify the characteristics of the FPN components in the form of prior information. Subsequently, a non-blind deconvolution strategy for thermal images was derived based on a comparative analysis of the gradient statistics of visible-band and infrared images. Finally, the constructed objective functions were successively minimized via the respective optimal algorithms. The convex problem of multiplicative and additive FPN was solved in the ADMM framework, and the spatially variant non-convex problem of deconvolution was optimized by the compositive utilization of IRLS and the conjugate gradient method. The efficacy of the proposed thermal image deconvolution framework was validated simulatively and experimentally by removing FPN and blur problems, and its superiority in performance was verified in terms of the PSNR, SSIM, Ro, and ERo.

## Figures and Tables

**Figure 1 sensors-23-03033-f001:**
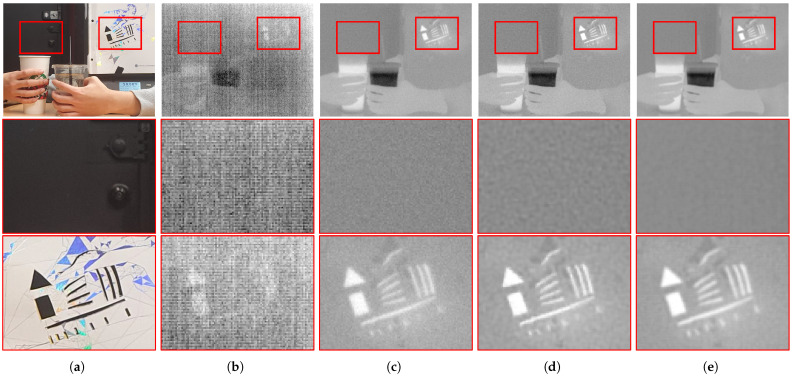
Comparative restored thermal images of an infrared camera. Thermal images commonly demonstrate quite worse qualities compared to visible-band images, so they require additional restoration processes. The conventional thermal image restoration framework consists of the FPN removal and deblur processes. The proposed framework jointly considers both problems in the deconvolution process. (**a**) Acquired visible-band image; (**b**) acquired thermal image; (**c**) FPN-removed image [14]; (**d**) deblurred image [11] after FPN removal process [14]; (**e**) restored image of the proposed framework.

**Figure 2 sensors-23-03033-f002:**
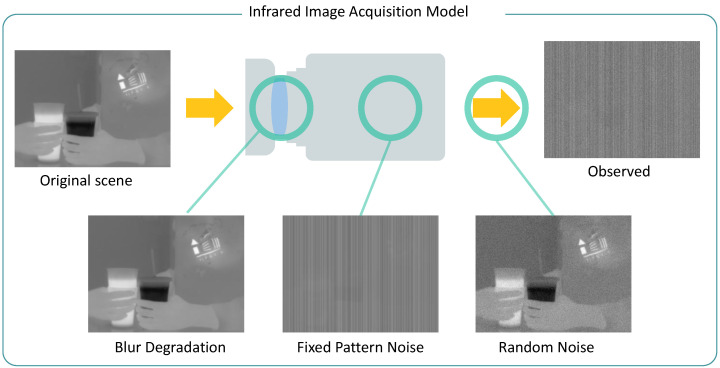
Framework of infrared image acquisition process. For the original scene, blur degradation occurs in the optical system, fixed pattern noise is generated by the responsivity of the sensor, and random noise is acquired during observation.

**Figure 3 sensors-23-03033-f003:**
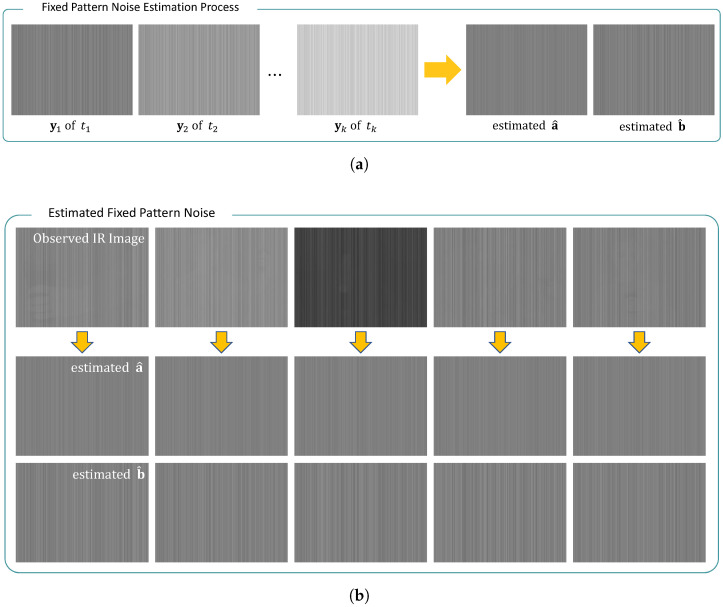
Materials of the proposed fixed pattern noise estimation algorithm. (**a**) Framework of the proposed FPN estimation algorithm using calibration data; (**b**) estimated FPN components of the proposed algorithm for five different infrared images.

**Figure 4 sensors-23-03033-f004:**
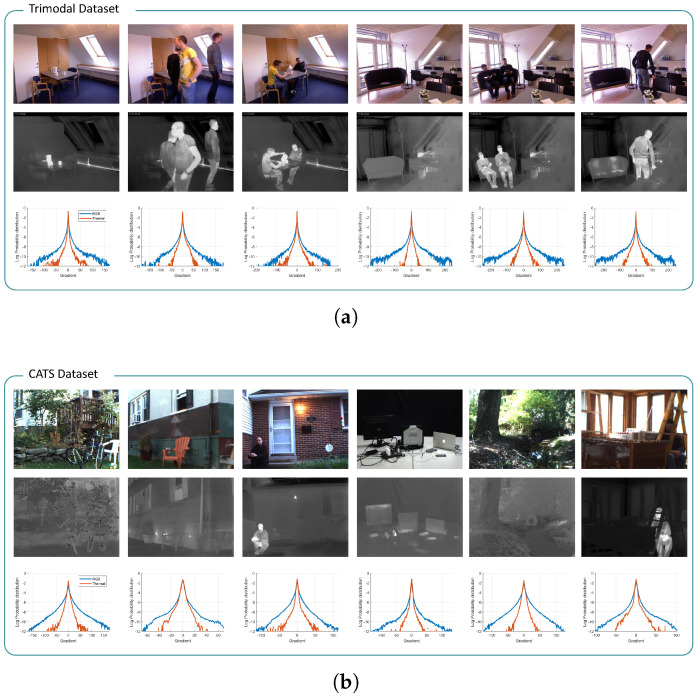
Visual comparison between visible-band images and infrared images for several datasets. Additionally, they are compared through their gradient distributions in the log scale. In the distributions, the blue and red lines represent the RGB images and thermal images, respectively. (**a**) Trimodal dataset [27]; (**b**) CATS dataset [28]; (**c**) FLIR dataset [29]; (**d**) SRIP dataset [13].

**Figure 5 sensors-23-03033-f005:**
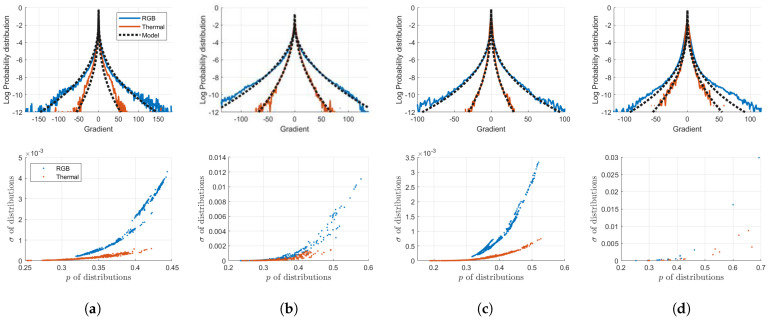
Examples of modeled gradient distributions of the individual RGB and thermal image dataset in Figure 4. The above gradient distributions are approximated to the GGD model with our previous statistical modeling algorithm. The graphs below exhibit the modeled parameters *p* (horizontal axis) and σ (vertical axis) for each image of the dataset. In the above distributions, the blue and the red lines denote RGB and thermal distributions, and the black dotted lines denote the statistical models describing them. In the graphs below, the blue and the red dots denote the statistical parameters of RGB and thermal images, respectively. (**a**) Trimodal dataset [27]; (**b**) CATS dataset [28]; (**c**) FLIR dataset [29]; (**d**) SRIP dataset [13].

**Figure 6 sensors-23-03033-f006:**
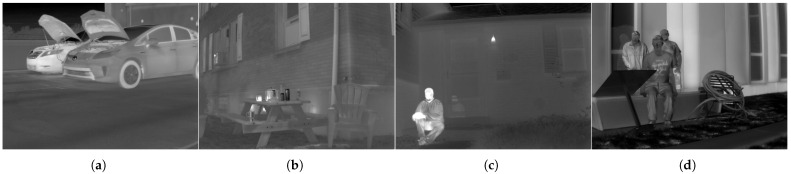
Test images selected from high-quality infrared image dataset [28]. Additional degradations are synthesized for the images according to the problems. (**a**) CARS; (**b**) COUNTRYARD; (**c**) HOUSE; (**d**) ISE.

**Figure 7 sensors-23-03033-f007:**
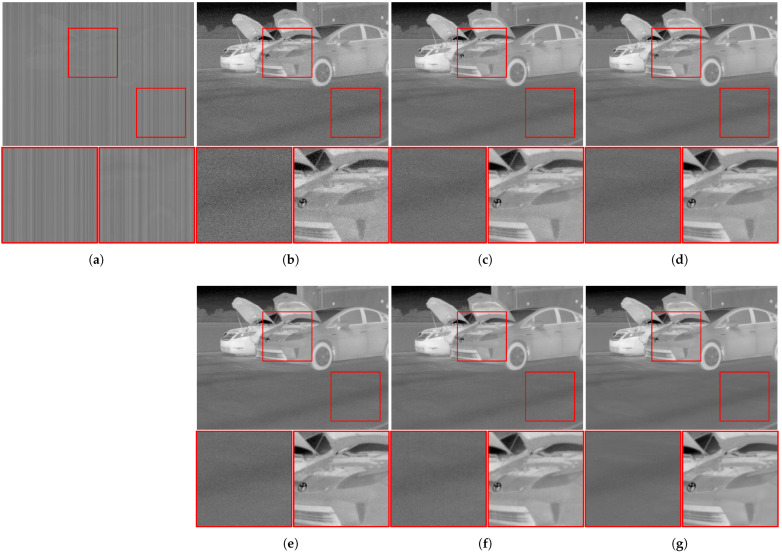
FPN-removed results of synthesized images restored by calibration-based FPN removal algorithms. (**a**) Degraded image; (**b**) two-point method [18]; (**c**) multi-point method [19]; (**d**) polynomial method [14]; (**e**) Proposed 1; (**f**) Proposed 2; (**g**) original image.

**Figure 8 sensors-23-03033-f008:**
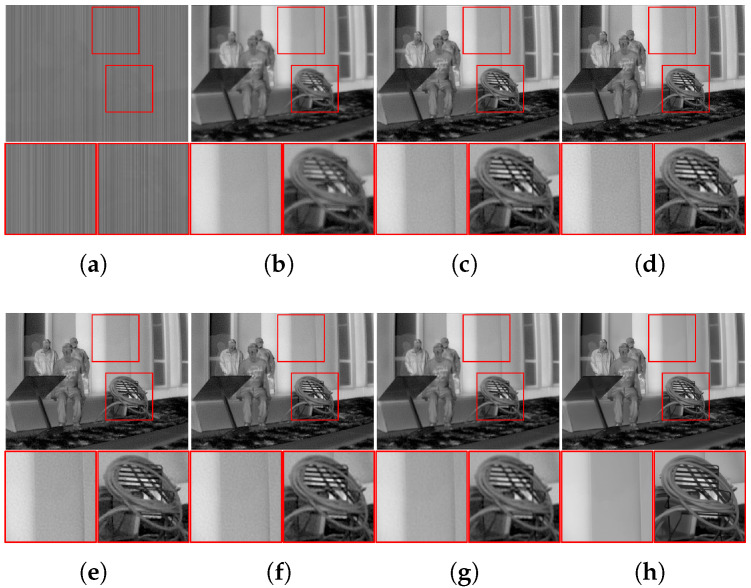
Thermal image deconvolution results of synthesized images restored by non-blind image deconvolution algorithms. (**a**) Degraded image; (**b**) initially denoised image [14]; (**c**) Levin et al. [6]; (**d**) Liming et al. [8]; (**e**) Jon et al. [11]; (**f**) Lee et al. [10]; (**g**) proposed; (**h**) original image.

**Figure 9 sensors-23-03033-f009:**
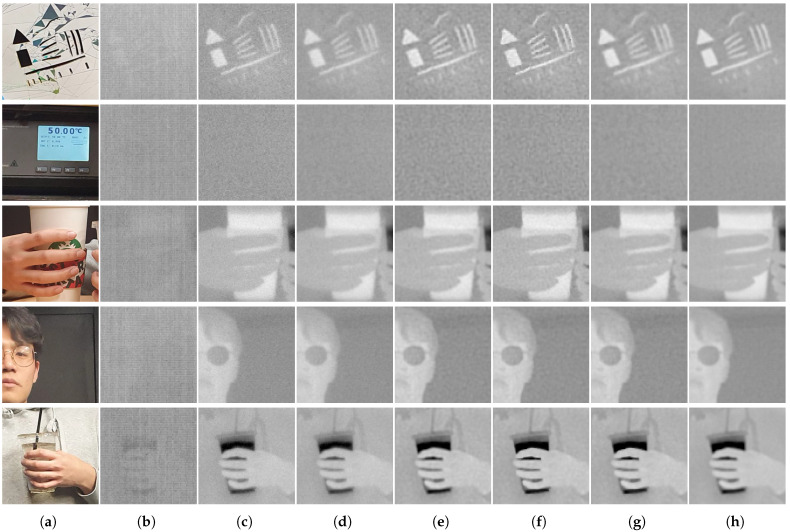
Thermal image deconvolution results of real-world infrared images restored by non-blind image deconvolution algorithms. (**a**) Visible-band image; (**b**) observed infrared image; (**c**) initially denoised image [14]; (**d**) Levin et al. [6]; (**e**) Liming et al. [8]; (**f**) Jon et al. [11]; (**g**) Lee et al. [10]; (**h**) proposed.

**Table 1 sensors-23-03033-t001:** Quantitative measurements of restored infrared image qualities of denoising algorithms using four quality assessments: PSNR, SSIM, Ro, and ERo.

Measurement	Figure 6	Degraded	Two-Point	Multi-Point	Polynomial	Proposed 1	Proposed 2
PSNR	CARS	16.418	27.992	33.189	34.297	35.885	39.748
	COUNT	22.833	35.169	40.200	41.315	42.863	44.616
	HOUSE	19.441	34.661	36.922	37.239	39.122	41.657
	ISE	13.490	31.844	35.545	36.068	37.850	40.062
SSIM	CARS	0.4861	0.4807	0.7329	0.7774	0.8330	0.9357
	COUNT	0.5411	0.8061	0.9286	0.9438	0.9602	0.9759
	HOUSE	0.5252	0.7715	0.8487	0.8575	0.9027	0.9461
	ISE	0.3668	0.7161	0.8493	0.8631	0.9046	0.9478
Ro	CARS	1.2146	0.2533	0.1493	0.1347	0.1157	0.0646
	COUNT	1.2231	0.1230	0.0811	0.0752	0.0682	0.0479
	HOUSE	1.2837	0.1556	0.1254	0.1224	0.1035	0.0679
	ISE	1.2092	0.1996	0.1449	0.1394	0.1214	0.0895
ERo	CARS	1.9039	3.0121	2.9257	2.8966	2.8312	2.4240
	COUNT	1.8813	2.9325	2.7921	2.7569	2.7004	2.4497
	HOUSE	1.9018	2.9968	2.9221	2.9150	2.8587	2.6381
	ISE	1.8886	2.8896	2.7512	2.7261	2.6291	2.3505

**Table 2 sensors-23-03033-t002:** Quantitative measurements of restored infrared image qualities of non-blind deconvolution algorithms using four quality assessments: PSNR, SSIM, Ro, and ERo.

Measurement	Figure 6	Degraded	Denoised [14]	Levin [6]	Liming [8]	Jon [11]	Lee [10]	Proposed
PSNR	CARS	16.356	29.576	32.549	33.256	34.240	33.473	33.488
	COUNT	22.823	33.592	37.454	37.271	38.163	37.867	38.058
	HOUSE	19.456	33.006	36.416	36.791	37.118	37.036	37.126
	ISE	13.437	29.470	34.371	34.154	35.215	35.050	35.091
SSIM	CARS	0.4804	0.6925	0.8655	0.8825	0.8756	0.8827	0.8962
	COUNT	0.5278	0.8604	0.9114	0.9015	0.9043	0.9151	0.9333
	HOUSE	0.5312	0.7827	0.9098	0.9050	0.8939	0.9076	0.9227
	ISE	0.3574	0.7484	0.8837	0.8538	0.8804	0.8877	0.9142
Ro	CARS	1.2262	0.1232	0.0555	0.0440	0.0514	0.0474	0.0406
	COUNT	1.2325	0.0588	0.0450	0.0463	0.0460	0.0430	0.0327
	HOUSE	1.2476	0.1091	0.0307	0.0354	0.0419	0.0385	0.0279
	ISE	1.2119	0.1217	0.0837	0.0945	0.0856	0.0832	0.0701
ERo	CARS	1.8624	3.0403	2.6509	1.9245	2.1347	2.2837	2.0028
	COUNT	1.8398	3.0057	2.3159	2.0497	2.2818	2.2158	1.9754
	HOUSE	1.8541	3.0411	2.2530	1.8918	2.0505	2.3234	1.9518
	ISE	1.8395	2.9929	2.0909	1.9028	2.1518	1.9719	1.7762

**Table 3 sensors-23-03033-t003:** Quantitative measurements of restored real infrared image qualities of non-blind deconvolution algorithms by following two quality assessments: Ro and ERo.

Dataset	Measurement	Observed	Denoised [14]	Levin [6]	Liming [8]	Jon [11]	Lee [10]	Proposed
SRIP	Ro	0.9240	0.0846	0.0586	0.0457	0.0439	0.0465	0.0291
	ERo	2.8831	2.9420	2.7735	2.0096	2.0193	2.3095	1.9874

## Data Availability

Not applicable.

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
