# Peer review of "Infrared Image Deconvolution Considering Fixed Pattern Noise"

_sensors, 2023, doi:10.3390/s23063033_

Round 1

Reviewer 1 Report

Review of “Infrared Image Deconvolution Considering Fixed Pattern Noise” by Lee et al. The manuscript proposes an infrared image deconvolution algorithm that considers both two problems (fixed pattern noise and blurring artifacts) in a single framework. The authors utilize the visual features of a column FPN to specify the characteristics of the FPN components, the gradient statistics of visible-band and infrared images to derive a non-blind deconvolution strategy. Subsequently, the objective functions are minimized using ADMM framework and the iteratively reweighted least squares algorithm. The superiority of the proposed thermal image deconvolution framework has been verified in terms of PSNR, SSIM, Ro, and ERo. I will recommend its publishing after some minor modifications.

Minor ones,

L13, delete the comma after the “both artifacts”.

L32, the complete spelling of ‘CMOS’ and ‘CCD’.

L241-242, Sentence grammatical error

Author Response

5, March, 2023

Ms. Adriana Serban

Assistant Editor

Sensors

Dear Editor:

We wish to re-submit the manuscript titled “Infrared Image Deconvolution Considering Fixed Pattern Noise.”  The manuscript ID is sensors-2265574.

We thank you and the reviewers for your thoughtful suggestions and insights. The manuscript has benefited from these insightful suggestions. I look forward to working with you and the reviewers to move this manuscript closer to publication in the Sensors.

The manuscript has been rechecked and the necessary changes have been made in accordance with the reviewers’ suggestions. The responses to all comments have been prepared and attached with the response letter and the revised changes have been incorporated into the manuscript. 

Thank you for your consideration. I look forward to hearing from you.

Sincerely,

Moon Gi Kang, Professor

School of Electrical and Electronic Engineering, Yonsei University,

Room C524, Building 123, 50 Yonsei-Ro, Seodaemun-gu, Seoul, 03722, Republic of Korea.

Phone: (822) 2123-4863,             Fax: (822) 312-4584

e-mail: mkang@yonsei.ac.kr

Reviewer 2 Report

The paper is not very much related to my narrow field of interest, that is applications of IR imaging rather than the denoising of source IR images supplied by contemporary IR detectors. However, the paper seems to be rather deep, interesting and generally well-written. In fact, there is one point (for the reviewer) to make the paper clearer.

In the examples, the authors use some source images of a very bad quality: the degraded images in Fig. 1b and many others like this. Were these images deliberately worsened to demonstrate efficiency of the proposed algorithm? I would say that image like in Fig. 1c is what we typically get with a medium quality IR imager. By other words, how the proposed algorithm would improve conventional IR images? Since night vision applications of IR imaging are vast, it seems that the suggested methodology may be useful to improve such images, for example, taken at a big distance, or under fog or smoke conditions. Is it possible to demonstrate the efficiency of the algorithm on such images?

The English of the paper is OK. There are few points to polish it. For example:

  • “Sensing thermal information” perhaps is not a good term. IR detectors do not sense information but radiation.
  • Instead of “past algorithm” would be better “the earlier-proposed algorithm”.

Author Response

(The authors gave the same response as above.)
